# Drilling Down into the Discourse Structure with LLMs for Long Document Question Answering

**Inderjeet Nair*[1], Shwetha Somasundaram*[2], Apoorv Saxena[2], Koustava Goswami[2]**

[1]University of Michigan, Ann Arbor, MI

[2]Adobe Research, India

inair@umich.edu

{shsomasu,apoorvs,koustavag}@adobe.com

## Abstract

We address the task of evidence retrieval for long document question answering, which involves locating relevant paragraphs within a document to answer a question. We aim to assess the applicability of large language models (LLMs) in the task of zero-shot long document evidence retrieval, owing to their unprecedented performance across various NLP tasks. However, currently the LLMs can consume limited context lengths as input, thus providing document chunks as inputs might overlook the global context while missing out on capturing the inter-segment dependencies. Moreover, directly feeding the large input sets can incur significant computational costs, particularly when processing the entire document (and potentially incurring monetary expenses with enterprise APIs like OpenAI's GPT variants). To address these challenges, we propose a suite of techniques that exploit the discourse structure commonly found in documents. By utilizing this structure, we create a condensed representation of the document, enabling a more comprehensive understanding and analysis of relationships between different parts. We retain 99.6% of the best zero-shot approach's performance, while processing only 26% of the total tokens used by the best approach in the information seeking evidence retrieval setup. We also show how our approach can be combined with *self-ask* reasoning agent to achieve best zero-shot performance in complex multi-hop question answering, just ≈ 4% short of zero-shot performance using gold evidence.

## 1 Introduction

Long Document Question Answering (LDQA) is a complex task that involves locating relevant evidence from lengthy documents to provide accurate answers to specific questions (Dasigi et al., 2021). LDQA is challenging for the following reasons -

a) Long documents often exceed the maximum token limit of existing transformer-based Pretrained Language Models (PLMs) (Devlin et al., 2019; Liu et al., 2019; Lewis et al., 2020; Raffel et al., 2020), posing a challenge in directly processing their content to extract pertinent information (Dong et al., 2023). b) The information required to answer a question is often dispersed across different sections or paragraphs within the document which may require sophisticated reasoning process to identify and extract the relevant information (Nie et al., 2022). c) Processing the entire document to find answers can be computationally expensive and inefficient (Dong et al., 2023).

One popular approach for LDQA is the retrieve-then-read method (Zheng et al., 2020; Gong et al., 2020; Nie et al., 2022; Ainslie et al., 2020, 2023), where relevant paragraphs are retrieved from the document to provide the answer. A major drawback of existing works is reliance on supervised fine-tuning for the evidence selection phase, exhibiting poor generalization on out-of-distribution data (Thakur et al., 2021).

Given the remarkable few-shot/zero-shot performance and enhanced generalization capabilities demonstrated by Large Language Models (LLMs) across various Natural Language Generation and Understanding tasks (Brown et al., 2020; Chen et al., 2021; Rae et al., 2022; Hoffmann et al., 2022; Chowdhery et al., 2022), we investigate the potential of leveraging these LLMs for zero-shot evidence retrieval. Notably, LLMs that have been instruction fine-tuned (Wei et al., 2022a; Chung et al., 2022) or trained using Reinforcement Learning with Human Feedback (Bai et al., 2022; Ouyang et al., 2022) exhibit exceptional generalization performance even on unseen tasks (Ouyang et al., 2022; Min et al., 2022; OpenAI, 2023). Thus, we explore the feasibility of utilizing LLMs for zero-shot evidence retrieval. However, LLMs, which are based on transformer architecture (Vaswani

---

* Equal contribution

[1] Work done at Adobe Research, India

et al., 2017), are limited by their context length and suffer from expensive inference times that increase quadratically with the number of tokens in the input. Additionally, utilizing enterprise LLM solutions such as OpenAI's gpt-3.5-turbo, text-davinci-003, gpt-4, etc.[1] to process an entire long document without optimizations would incur significant monetary costs. This highlights the need for an LLM-based evidence retrieval solution that can achieve faster and more cost-effective inference by selectively processing relevant portions of the document, without compromising downstream performance.

To overcome these challenges, we harness the inherent discourse structure commonly present in long documents. This structure encompasses the organization of topics, semantic segments, and information flow, enabling effective information search and knowledge acquisition for question answering. (Guthrie et al., 1991; Meyer et al., 1980; Taylor and Beach, 1984; Cao and Wang, 2022; Dong et al., 2023; Nair et al., 2023). Utilizing this valuable structure, we construct a condensed representation of the document by replacing the content within each section with a corresponding summary. This condensed representation is then fed to the LLM, enabling efficient processing of tokens while allowing the model to comprehensively analyze the entire input context for identifying relevant sections. Thereafter, the content within each relevant section is further processed by the LLM for fine-grained evidence retrieval. We call our proposed approach $D^3$ (Drilling Down into the Discourse) due to the nature of the solution described above.

Our approach undergoes evaluation in two distinct settings: Information Seeking and Multi-hop Reasoning in Question Answering. In the information seeking experiments, our approach retains the best zero-shot state-of-the-art (SoTA) results, while only utilizing 26% of the tokens employed by the SoTA approach. Additionally, we examine the robustness of our model across various document lengths and analyze the number of tokens required and latency for different zero-shot approaches. Moreover, we explore the integration of our approach with other zero-shot techniques within an agent framework designed to break down intricate queries into a sequence of simpler follow-up queries.

[1] https://openai.com/pricing

## 2 Related Work

### 2.1 LLMs in Retrieve-Then-Read Approaches

The retrieve-then-read (Green Jr et al., 1961; Chen et al., 2017; Wang et al., 2018; Das et al., 2019; Guu et al., 2020) approach is a widely adopted technique in open-domain (Voorhees et al., 1999; Dunn et al., 2017; Joshi et al., 2017; Zhu et al., 2021), multi-document question answering (Yang et al., 2018; Perez et al., 2020; Ferguson et al., 2020) and long-document question answering (Pereira et al., 2023). In this approach, LLMs are utilized specifically for the reader component, which generates responses based on the relevant fragments retrieved by the retriever (Pereira et al., 2023). Although LLMs have been utilized as decision-making agents in browser interactions for document retrieval (Nakano et al., 2022), their direct application for fine-grained evidence retrieval has not been extensively explored to the best of our knowledge. On that front, our paper is the first to evaluate the applicability of LLMs for evidence retrieval.

### 2.2 Chaining LLMs Runs for Question Answering

Chaining in LLMs refers to the task of breaking complex overarching task into a sequence of fine-grained targeted sub-tasks where the information generated by a particular run of the sequence is passed to the subsequent runs (Wu et al., 2022a,b). This allows for realizing powerful machine learning applications without requiring any changes to the model architecture (Tan et al., 2021; Betz et al., 2021; Reynolds and McDonell, 2021). LangChain has implemented procedures using chaining for evidence retrieval and question answering in long documents. They employ three chaining variants (map-reduce, map-rerank, and refine)[2], which processes document chunks individually and aggregate the information from each chunk to derive the final answer. This implementation, however, processes the entire document input resulting in significant compute and monetary cost.

### 2.3 Evidence Retrieval for LDQA

Prior evidence retrieval approaches typically employ following two mechanims which are trained by supervised fine-tuning - local processing to handle individual document chunks with occasional information flow between them (Gong et al., 2020)

[2] https://python.langchain.com/docs/modules/chains/document/

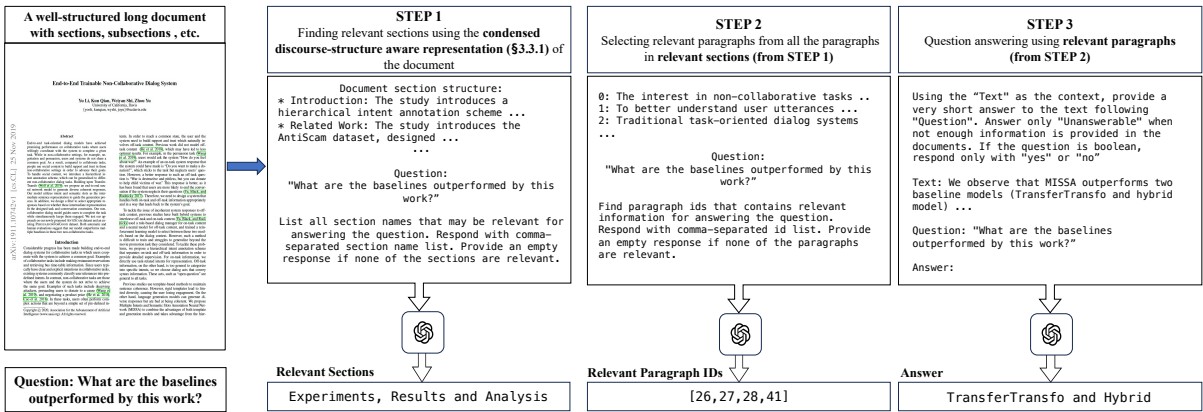

Figure 1: **An illustration of the end-end pipeline of $D^3$.** Given a question and a long document with discourse structure that indicates sections, subsections etc., we first identify sections that are relevant for answering the question. Following this step we select relevant paragraphs from the paragraphs in the relevant sections. In the final step, we pass these relevant paragraphs to an LLM for question answering.

and global processing to aggregate the information from each chunk to identify relevant paragraphs (Zheng et al., 2020; Ainslie et al., 2020; Nie et al., 2022; Ainslie et al., 2023). Inspired by this strategy, our method represent each section using its corresponding summary in a local processing step and, in the global processing mechanism, we utilize a suitable verbalizer to concatenate the summaries from each section.

## 3 $D^3$: Drilling Down into the Discourse

### 3.1 Problem Formulation

In LDQA, a question $\mathbf{q}$ is asked for a document $\mathbf{D} = [p_1, p_2, \ldots, p_n]$, where $p_i(1 \le i \le n)$ is the $i^{th}$ paragraph in the natural reading order of $\mathbf{D}$. The task of LDQA is to retrieve a set of relevant paragraphs $\mathbf{E_q} \subseteq \mathbf{D}$ and generate a free-form answer $\mathbf{a}$ based on $\mathbf{q}$ and $\mathbf{D}$ (Dasigi et al., 2021; Nie et al., 2022). Due to the length of the documents, often exceeding 5K tokens, we employ the retrieve-then-read strategy. This approach involves first determining $\mathbf{E_q}$ and subsequently generating $\mathbf{a}$ using only $\mathbf{q}$ and $\mathbf{E_q}$.

### 3.2 Motivation

The cognitive strategy employed by humans to search for relevant information from a document entails a systematic approach of first categorizing the information within the document to determine relevant coarse segments and then conducting a deeper analysis of the relevant categories to extract fine-grained segments (Guthrie and Kirsch, 1987; Guthrie et al., 1991; Guthrie and Mosenthal, 1987). Long documents often possess a well-structured

discourse that categorizes information coherently based on topical similarity (Cao and Wang, 2022; Nair et al., 2023; Dong et al., 2023). This inherent discourse structure serves as a valuable framework, enabling effective categorization of information and facilitating a clear and logical flow of ideas within the document.

Drawing from these insights, we posit that encapsulating the essence of a section through its name and content summary would yield valuable cues in determining its relevance for answering specific questions. Thereafter, we emulate the above-described cognitive process by fine-grained analysis of relevant sections to extract evidence paragraphs. This methodology offers three key advantages:
**(1)** By condensing each section with its name and summary, we can effectively reduce the document's token count, enabling LLMs to analyze the entire context and make accurate inferences.
**(2)** By efficiently filtering out irrelevant sections in the initial stage, our method reduces the number of tokens processed.
**(3)** Our method is applicable for any instruction-following LLMs (Ouyang et al., 2022; Min et al., 2022; OpenAI, 2023), enabling zero-shot application without the need for architectural modifications.

### 3.3 Methodology

Instead of representing a document $\mathbf{D}$ as a ordered set of constituent paragraphs, we repre-

sent $\mathbf{D} = [S_1, S_2, \ldots, S_k]^3$, where $S_i(1 \leq i \leq k)$ denotes $i^{th}$ section, such that, $\texttt{name}(S_i)$ and $\texttt{paragraphs}(S_i)$ denotes its name / heading and the list of constituent paragraphs respectively ($\texttt{paragraphs}(S_i) = [p_{i,j}]_{j=1}^{|S_i|}$ where $|S_i|$ denotes number of constituent paragraphs). Note that, $\sum_{i=1}^{k} |S_i| = n$. Inspired by the cognitive process of knowledge acquisition / information search for question answering, our approach first finds the relevant sections that may answer the question and then, analyses the paragraphs from the relevant sections for fine-grained evidence paragraph retrieval. (Figure 1)

### 3.3.1 Finding Relevant Sections

The crux of this step is to represent the content in each section $S_i$ by the summary of $\texttt{paragraphs}(S_i)$. Summarization (El-Kassas et al., 2021) refers to the task of generating a concise summary for a given input that captures its main idea within a limited number of tokens, effectively conveying its topical essence. We denote the summarization operation by $\mathcal{S}$, for which we have used $\texttt{bart-large}$ (Lewis et al., 2020) fine-tuned over CNN/Daily-Mail Corpus (Nallapati et al., 2016). Thereafter, we represent the entire document as follows:

```
* Section: name(S₁)
  𝒮(paragraphs(S₁))
* Section: name(S₂)
  𝒮(paragraphs(S₂))
        ...
* Section: name(Sₖ)
  𝒮(paragraphs(Sₖ))
```

An instruction is passed to an LLM involving the above representation to identify all the sections that are relevant to $\mathbf{q}$. Due to this condensed representation, the LLM can process the entire document context enabling comprehensive analysis of long range dependencies for accurate inference. Let the set of sections identified as relevant be denoted by $\mathbf{R_q} \subseteq \mathbf{D}$.

### 3.3.2 Fine-Grained Evidence Retrieval

The objective of this step is to infer the set of relevant paragraphs from $\mathbf{R_q}$. Here, we explain mul-

tiple zero-shot strategies to achieve this step. We, first, obtain a set of all paragraphs $\mathbf{P_q}$ associated with $\mathbf{R_q}$.

$$\mathbf{P_q} = \bigcup_{S \in \mathbf{R_q}} \texttt{paragraphs}(S)$$

Thereafter, one of the following strategy can be employed for fine-grained retrieval:

1. MONOT5: This employs MonoT5 (Nogueira et al., 2020), which a sequence-to-sequence model trained over the task of Document Reranking (Nguyen et al., 2016), to select the most relevant paragraphs from $\mathbf{P_q}$.

2. BASE: Each paragraph from $\mathbf{P_q}$ is marked with an identifier and then, these identifier annotated paragraphs are concatenated with a newline separator. Thereafter, we prompt the LLM in a zero-shot manner to generate all paragraph identifiers whose corresponding paragraph is relevant to $\mathbf{q}$. If the number of paragraphs in $\mathbf{P_q}$ exceeds the maximum context length of LLM, we make multiple LLM calls. In each call, we fit the maximum number of paragraphs that can fit into the context length, ensuring that paragraphs are not 'chopped'.

3. HIERBASE: In our approach, we adopt a two-step process to capture the essence of each paragraph. Firstly, we represent paragraphs using their corresponding summaries obtained through $\mathcal{S}$. Following that, we employ the BASE strategy to identify potentially relevant candidates. In the next stage, we apply the BASE technique once again, this time considering the original content of the paragraphs, to pinpoint the most relevant ones.

In our experimental evaluation, we also explore the effectiveness of chaining these strategies in succession. One such method, called BASE + MONOT5, combines the BASE strategy with MONOT5. This approach initially identifies a relevant set of paragraphs using the BASE strategy and subsequently employs MONOT5 to refine the selection further, retaining only the most relevant ones from the initially identified set.

For most of the experiments presented in the upcoming sections, we use $\texttt{bart-large}$ (Lewis et al., 2020) trained over the CNN/Daily-Mail Corpus (Nallapati et al., 2016) for $\mathcal{S}$. For retrieval

---

[3] In practice, documents often have a hierarchical discourse structure, consisting of multiple levels of sections (Nair et al., 2023). To handle this, we can flatten the structure using a pre-order traversal approach. When verbalizing a specific section, we concatenate the names of all sections along the path from the root node to that particular node in the discourse structure. This flattening process allows us to represent the document as a list of sections while considering the hierarchical relationships among sections.

| Approach | Answering Performance | | | | | Evidence $F_1$ | Tokens Processed | API Calls |
|---|---|---|---|---|---|---|---|---|
| | Extractive | Abstractive | Yes/No | Unanswerable | Overall | | | |
| HUMAN (Dasigi et al., 2021) | 58.92 | 39.71 | 78.98 | 69.44 | 60.92 | 71.62 | - | - |
| CGSN (Nie et al., 2022) | 34.75 | 14.39 | 68.14 | 71.84 | 39.44 | 53.98 | - | - |
| LED* (Nie et al., 2022) | 52.41 | 23.44 | 76.96 | 77.91 | 52.87 | - | - | - |
| gpt-3.5-turbo* | 54.86 | 27.74 | 81.50 | 95.76 | 57.99 | - | - | - |
| ZERO-SHOT / UNSUPERVISED METHODS | | | | | | | | |
| MONOT5 (Nogueira et al., 2020) | 42.84 | 25.84 | 82.23 | 69.09 | 47.21 | 34.23 | - | - |
| DPR (Karpukhin et al., 2020) | 31.58 | 18.57 | 78.46 | 84.33 | 42.11 | 19.32 | - | - |
| cross-encoder-ms-marco-MiniLM-L-12-v2 | 38.69 | 23.25 | 78.04 | 71.42 | 43.48 | 30.76 | - | - |
| PARAGRAPH | 45.20 | 26.02 | 76.13 | 72.56 | 47.92 | 32.02 | 8519.37 | 47.24 |
| CHUNK | 45.96 | **29.61** | **84.30** | 65.57 | 49.00 | 35.59 | 5411.44 | 2.44 |
| MAP-REDUCE | 21.37 | 19.65 | 76.47 | **90.28** | 39.26 | 12.84 | 12730.13 | 48.24 |
| MAP-REDUCE OPTIMIZED | **47.45** | 26.05 | 82.79 | 71.32 | **50.13** | **50.11** | 7491.97 | 3.69 |
| $\mathbf{D}^3$-BASE | 42.90 | 23.65 | 74.35 | 79.61 | 47.45 | 49.92 | **1980.94** | **1.99** |

Table 1: **Comparison of various zero-shot approaches against SoTA methods for QASPER dataset.** The simplest algorithm from $\mathbf{D}^3$ family yields competitive value across several metrics while being zero-shot and requiring least number of tokens. *: Inference obtained using gold evidence.

and question answering, we utilize the highly capable gpt-3.5-turbo model, known for its remarkable performance across a wide range of NLP tasks, all while being more cost-effective (Ye et al., 2023) when compared against text-davinci-003. To identify the most relevant sections§3.3.1, we prompt the LLM with the following instruction:

```
Document section structure:
{Condensed representation described in §3.3.1}
Question:
{q}
List all section names that may be relevant for
answering the question. Respond with
comma-separated section name list. Provide an
empty response if none of the sections are
relevant.
```

For the BASE strategy described in §3.3.2, we employ the following prompt:

```
{Paragraphs annotated with identifier (§3.3.2)}
Question:
{q}
Find paragraph ids that contains relevant
information for answering the question. Respond
with comma-separated id list. Provide an empty
response if none of the paragraphs are relevant.
```

## 4 Baselines:

We consider the following zero-shot approaches for performance comparison:
**(1)** MONOT5: In this approach, MONOT5 is directly applied to re-rank the paragraphs of $\mathbf{D} = [p_1, p_2, \ldots, p_n]$ based on $\mathbf{q}$.
**(2)** DPR: Dense Passage Retrieval (DPR) (Karpukhin et al., 2020) is a retrieval approach that leverages dense representations. This method utilizes a bi-encoder architecture to generate embeddings for both the documents and the query independently which are used for finding the most relevant documents.

**(3)** cross-encoder-ms-marco-MiniLM-L-12-v2: This model is a cross-encoder reranker which employs Siamese learning and BERT-like architecture. This model is offered in the sentence-transformers (Reimers and Gurevych, 2019) library. While the library provides many different model checkpoints, we chose cross-encoder-ms-marco-MiniLM-L-12-v2 as it yielded highest $F_1$ score for evidence retrieval.
**(4)** PARAGRAPH: Every paragraph in $\mathbf{D}$ is processed by the LLM independently to assess its relevance to $\mathbf{q}$ through a boolean prompt.
**(5)** CHUNK: The document is partitioned into consecutive fragments, aiming to accommodate as many paragraphs as possible within a predefined token limit called as chunk size (3500). Subsequently, BASE is applied for each fragment.
**(6)** MAP-REDUCE: This approach, widely adopted in the literature, involves evaluating the relevance of each paragraph to $\mathbf{q}$ and subsequently processing the relevant paragraphs together in a single call to the LLM. However, we observed that using LangChain's implementation directly led to subpar performance in terms of evidence retrieval $F_1$-score and inference cost. This can be attributed to the incompatibility of the prompt with the target domain, resulting in significant performance degradation due to gpt-3.5-turbo's high sensitivity to the prompt (Ye et al., 2023).
**(7)** MAP-REDUCE OPTIMIZED: For better alignment with our target task, we made crucial modifications to the original implementation. Building upon the observation that CHUNK outperforms PARAGRAPH in terms of performance, we decided to process document chunks instead of individual paragraphs using the BASE tech-

nique. Following the initial stage, where relevant paragraphs are identified, we concatenate them and subject them to the same strategy (BASE) for further processing.

# 5 Experiments and Results

We mainly assess the applicability of our method in two scenarios: **(1)** §5.1: Information-seeking setting (Dasigi et al., 2021) and **(2)** §5.1: Multi-hop Reasoning in Question Answering (Yang et al., 2018). Thereafter, we conduct an extensive analysis of our approach's performance across various configurations, as discussed in §5.3.1, and examine its effectiveness in different document length categories, as outlined in §5.3.2. Thereafter, we justify the need for the inclusion of global context modeling in §5.3.3, and highlight the significance of incorporating discourse information in §5.3.4. Finally, §5.3.5 we compare our methods performance with the best performing zero-shot approach for different categories and identify scope for improvement. In our analyses, we will also include insights into the inference cost and latency associated with utilizing large language models (LLMs) for the **evidence retrieval stage**. This will encompass the monetary cost and the processing efficiency measured in terms of the number of tokens processed and the number of LLM inferences. It is important to note that these measurements pertain exclusively to LLMs and do not encompass smaller fine-tuned models. Unless otherwise specified, we would be using BASE approach for fine-grained retrieval and the overall approach would be called $\mathbf{D}^3$-BASE. Due to the monetary cost associated with `gpt`-based LLMs, we experiment with 150 randomly sampled documents for all the experiments.

## 5.1 Performance for Information-Seeking Setting

We assess the performance of the models on the QASPER dataset (Dasigi et al., 2021), which comprises information-seeking questions designed for lengthy research papers. The dataset includes a set of ground truth evidence paragraphs and answers. The questions are categorized as extractive, abstractive, yes/no, and unanswerable, and our proposed method must accurately discern the question's intent to generate a relevant response.

Table 1 presents the results of various approaches, including a fine-tuned state-of-the-art

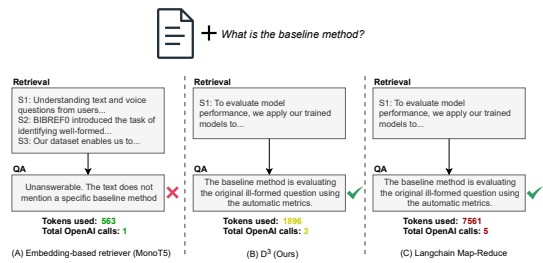

Figure 2: **Qualitative comparison of $\mathbf{D}^3$-BASE with publicly available Zero-shot Approaches such as MONOT5 and Langchain's MAP-REDUCE**: "Tokens used" refers to the total number of tokens processed by the evidence retrieval and question answering stage to generate the final answer. Similarly "Total OpenAI Calls" also computes the number of API calls over both the tasks.

(SoTA) model (Nie et al., 2022), fine-tuned LED model evaluated on gold evidence, and zero-shot `gpt-3.5-turbo` model evaluated on gold evidence. Among them, our simplest approach, $\mathbf{D}^3$-BASE, achieves competitive performance in terms of Evidence $F_1$ score. Notably, it retains 99.6% of the performance of the best zero-shot approach, MAP-REDUCE OPTIMIZED, while processing only 26% of the tokens required by the latter.

The original implementation of MAP-REDUCE suffers from two main limitations. Firstly, it processes each paragraph independently, overlooking the effectiveness of chunk-level processing over paragraph-level processing. Secondly, it employs suboptimal few-shot prompting to retrieve relevant sources, resulting in increased processing costs and poor performance when the few-shot prompting is not well-aligned with the domain. Due to these significant processing costs and the underperformance of PARAGRAPH and MAP-REDUCE, we exclude their evaluations from subsequent analyses. Similarly, among the non-LLM baselines, we exclude the evaluations of DPR and `cross-encoder-ms-marco-MiniLM-L-12-v2` due to their poor performance.

While it is possible to enhance the performance of our approach, $\mathbf{D}^3$, with alternative configurations (as shown in Section 5.3.1), it remains an excellent choice for rapid testing in different domains. Its cost-effectiveness and minimal latency in terms of API calls make it highly suitable for such purposes.

| Approach | Evidence $F_1$ | Answer $F_1$ | Tokens Processed | API Calls |
|---|---|---|---|---|
| CGSN (Nie et al., 2022) | 92.02 | 57.80 | - | - |
| LED* (Nie et al., 2022) | - | 58.94 | | |
| gpt-3.5-turbo* | - | 47.01 | - | - |
| ZERO-SHOT APPROACHES | | | | |
| MONOT5 | **62.33** | 37.91 | - | - |
| CHUNK | 40.79 | 36.81 | 6702.78 | 2.45 |
| MAP-REDUCE OPTIMIZED | 39.52 | **37.93** | 8329.81 | 3.58 |
| $\mathbf{D}^3$-BASE | 31.05 | 23.55 | **3141.29** | **2.10** |
| ZERO-SHOT APPROACHES AUGMENTED WITH SELF-ASK | | | | |
| MONOT5 | **33.56** | 40.36 | **719.54** | **1.62** |
| CHUNK | 20.66 | 39.59 | 9822.49 | 5.0 |
| MAP-REDUCE OPTIMIZED | 30.19 | 38.32 | 12253.42 | 6.83 |
| $\mathbf{D}^3$-BASE | 26.87 | **43.45** | 5376.29 | 5.17 |

Table 2: **Comparison of various zero-shot approaches for HOTPOTQA-Doc Dataset.** While directly applying $\mathbf{D}^3$-BASE leads to poor performance, combining this with *self-ask* prompting methodology yields best performance while processing least number of tokens when compared against other zero-shot LLM-based methods. *: Inference obtained using gold evidence.

## 5.2 Performance for Questions Requiring Multi-Hop Reasoning

Here, we use HOTPOTQA-Doc dataset (Yang et al., 2018; Nie et al., 2022), where the objective is to answer a complex query involving multi-hop reasoning given two long documents. We have investigated the performance of different zero-shot approaches using two schemes: **(a) Direct Processing:** Queries are directly fed to the Zero-Shot retrievers to get the relevant evidences. **(b)** *self-ask* **based Processing:** By leveraging the power of elicitive prompting (Yao et al., 2022; Press et al., 2022; Wei et al., 2022b), we employ the technique of *self-ask* (Press et al., 2022). This approach entails decomposing a complex query into a series of simpler questions, which collectively form the basis for the final answer. Through iterative questioning, the agent analyzes prior answers and previously posed questions to generate subsequent inquiries. Leveraging the zero-shot retrieval approach, the agent obtains relevant answers for each question.

The results for this experiment are tabulated at Table 2. We note the following observations:

- **Evidence $F_1$ poorly correlated with Answer $F_1$**: Considering same question answering model were used (gpt-3.5-turbo) for each of the zero-shot approaches, we find that the answer performance of MAP-REDUCE OPTIMIZED aligns with that of MONOT5, albeit with noticeably lower evidence retrieval efficacy. It's worth noting that during dataset construction, only the paragraphs utilized for gen-

erating answers were preserved, which does not imply the irrelevance of other paragraphs in addressing the question. This observation highlights the presence of this artifact.

- **Augmenting with *self-ask* boosts performance**: This highlights the fact that zero-shot retrievers are better positioned to retrieve fragments for simpler queries and *self-ask* effectively uses them to get better performance. In fact, our approach $\mathbf{D}^3$ − BASE is very close in performance to zero-shot question answering with gold evidence.

## 5.3 Ablations & Analyses

### 5.3.1 Performance of different configurations of $\mathbf{D}^3$

We investigated several configurations to determine possible directions to improve the performance. Following explorations were performed (Table 3):

- **Variations in fine-grained retrieval**: Although employing MonoT5 can reduce inference costs, it also adversely affects evidence retrieval performance, as evidenced by the table. Conversely, $\mathbf{D}^3$−HIERBASE demonstrates a enhancement in performance with only a marginal increase in inference cost.

- **Using LLMs for summarization**: We replaced fine-tuned summarizer with enterprise LLMs such as gpt-3.5-turbo[4] and text-davinci-003 (Ouyang et al., 2022) and an open-source LLM, vicuna-13b (Chiang et al., 2023). While the performance increased along several metrics, there is additional processing cost to get the condensed representation of the document (very minimal when smaller bart-large based summarizer was used).

- **Exploring alternative LLMs for retrieval**: In this analysis, we observe a decline in performance when utilizing other LLMs, highlighting the superiority of gpt-3.5-turbo as the optimal choice for retrieval tasks.

- **Investigating alternative LLMs for question answering**: We observe a performance boost when employing text-davinci-003. However, it is important to consider the higher monetary cost associated with using this API compared to gpt-3.5-turbo.

---

[4]https://openai.com/blog/chatgpt

| Approach | Answering Performance | | | | | Evidence $F_1$ | Tokens Processed | API Calls |
|---|---|---|---|---|---|---|---|---|
| | Extractive | Abstractive | Yes/No | Unanswerable | Overall | | | |
| *OUR PRIMARY APPROACH* | | | | | | | | |
| $\mathbf{D^3}$-BASE | 42.90 | 23.65 | 74.35 | 79.61 | 47.45 | 49.92 | 1980.94 | 1.99 |
| *VARIATIONS IN FINE-GRAINED EVIDENCE RETRIEVAL* | | | | | | | | |
| MONOT5 | 34.86 | 20.47 | 67.59 | **88.64** | 43.33 | 32.73 | **844.09** | **1.0** |
| MONOT5+BASE | 39.61 | 22.31 | 74.32 | 87.35 | 47.19 | 44.39 | 1520.35 | 1.95 |
| BASE+MONOT5 | 39.62 | 23.66 | **74.54** | 82.33 | 46.42 | 40.23 | 1980.94 | 1.99 |
| HIERBASE | **45.48** | **24.14** | 71.55 | 86.18 | **49.48** | **50.09** | 2125.67 | 2.85 |
| *REPLACING FINE-TUNED SUMMARIZATION MODEL WITH INSTRUCTION ALIGNED LLM* | | | | | | | | |
| gpt-3.5-turbo | **43.89** | 27.31 | **79.81** | 75.55 | **49.28** | **51.19** | **2106.57** | **2.0** |
| text-davinci-003 | 43.04 | 27.04 | 79.28 | 76.32 | 48.61 | 50.37 | 2239.97 | 2.03 |
| vicuna-13b | 43.03 | **27.70** | 75.16 | **79.55** | 49.09 | 50.09 | 2208.63 | 2.14 |
| *VARYING LLMs FOR RETRIEVAL* | | | | | | | | |
| text-davinci-003 | **41.45** | **24.12** | **79.08** | 77.72 | **47.11** | **37.53** | 2673.26 | 2.0 |
| vicuna-13b | 23.35 | 15.38 | 74.07 | **86.88** | 36.52 | 28.10 | **1810.48** | **1.85** |
| *VARYING LLMs FOR QUESTION ANSWERING* | | | | | | | | |
| text-davinci-003 | **49.99** | 20.33 | **77.86** | **90.82** | **52.51** | 49.92 | 1980.94 | 1.99 |
| vicuna-13b | 31.71 | **21.34** | 62.29 | 60.43 | 37.06 | 49.92 | 1980.94 | 1.99 |

Table 3: **Performance for different $\mathbf{D^3}$ configurations for QASPER dataset.**

### 5.3.2 Performance across Different Document Length categories

We divided the test set into different categories based on their respective lengths. We notice the advantage of our method along three aspects:

- **Evidence $F_1$**: Our approach consistently achieves competitive performance in evidence retrieval across various document length categories (Figure 3 (a)).

- **Evidence Retrieval Cost**: Our approach significantly reduces the number of processed tokens compared to other methods in all document length categories (Figure 3 (b)). This cost-efficient characteristic makes it an excellent choice for minimizing both inference and monetary costs, regardless of the document's length .

- **Latency**: Irrespective of the document's length, our approach maintains a minimal latency by making approximately 2 API calls to the LLM (Figure 3 (c)). This efficient performance further highlights its desirability and suitability for various applications.

### 5.3.3 Need for Global Context Modeling

In this section, we investigate the effect of changing the chunk size of the baseline CHUNK in terms of evidence precision, recall and $F_1$-score. As we see from Figure 4, the precision and $F_1$-score increases at the cost of modest decrease in recall as the chunk-size increasing. This observation underscores the fact that larger chunk sizes enable the model to capture longer-range relationships and contextual information, resulting in improved performance. By processing the entire document in a single pass, $\mathbf{D^3}$ benefits from accessing the global context and long-range relationships, leading to enhanced performance.

### 5.3.4 Role of Discourse Headings and Sub-headings

| Approach | Evidence $F_1$ | Tokens Processed | API Calls |
|---|---|---|---|
| $\mathbf{D^3}$-BASE | 49.92 | 1980.94 | 1.99 |
| $\mathbf{D^3}$-BASE $\times$ | 43.11 | 2183.55 | 1.88 |

Table 4: **Performance of different methods with and without section information (denoted by ×) for QASPER dataset.**

To assess the importance of section headings / sub-headings in the discourse structure for retrieval, we replaced them with randomly initialized Universally Unique Identifiers (UUIDs) and test evidence retrieval performance over the QASPER dataset (Table 4). The significant decrease in the performance shows that section headings / sub-headings are crucial in conveying the topical essence of a section, which is needed for accurate retrieval.

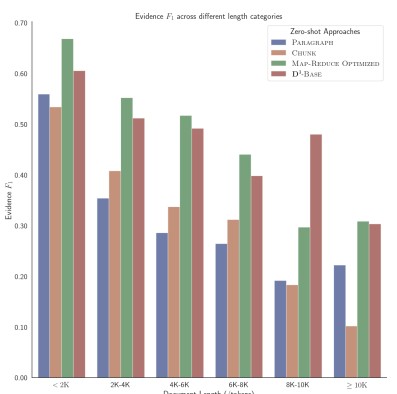
(a) Evidence $F_1$ across different document length categories

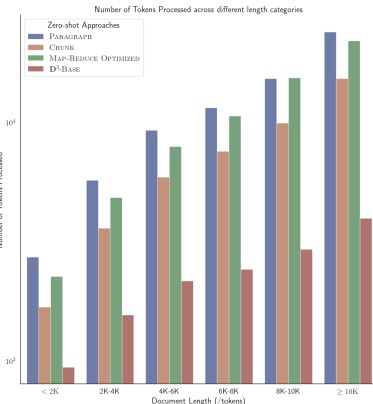
(b) Tokens Processed for retrieval across different document length categories

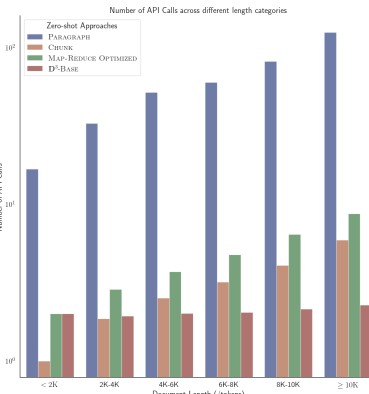
(c) API calls for retrieval across different document length categories

Figure 3: **Analysing the performance of different approaches across different length categories along three metrics for QASPER Dataset**

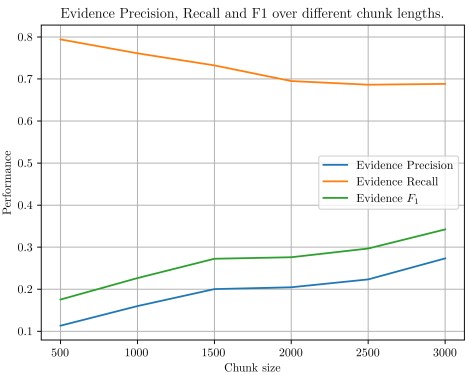

Figure 4: **Evidence Precision, Recall and $F_1$ over different chunk lengths for QASPER Dataset**

### 5.3.5 Evidence Retrieval Performance over Question Categories

In this section, we compare $D^3$-BASE with the best performing zero-shot baseline MAP-REDUCE OPTIMIZED (MRO) over different question categories in QASPER to identify differences in model performance (Table 5). While our approach is precise in identifying the evidence paragraphs, it occasionally falls short in identifying all pertinent evidence (i.e. lower recall). This indicates that representing a section with a summary leads to loss of information and the LLM may miscategorize its relevancy to a question due to this loss of information. Designing an effective strategy to prevent loss of vital information for a particular question may be a future research direction.

| Question Category | Evidence Precision ($D^3$ / MRO) | Evidence Recall ($D^3$ / MRO) | Evidence $F_1$ ($D^3$ / MRO) |
|---|---|---|---|
| Extractive | 52.63 / 52.19 | 71.9 / 81.49 | 56.39 / 57.46 |
| Abstractive | 41.25 / 45.1 | 52.51 / 71.57 | 41.78 / 50.12 |
| Yes/No | 43.6 / 38.79 | 59.6 / 55.31 | 45.53 / 40.18 |
| Unanswerable | 42.94 / 25.74 | 44.4 / 25.46 | 43.43 / 25.10 |

Table 5: **Comparison of evidence retrieval performance of $D^3$-BASE with the best performing zero-shot baseline MAP-REDUCE OPTIMIZED (MRO)**

## 6 Conclusion

We demonstrated a zero-shot approach for evidence retrieval which leverages the discourse structure of the document for information categorization which not only yielded competitive performance for information seeking and question answering with multi-hop reasoning setting, but also processed lowest number of tokens resulting in significant compute and cost savings. This approach demonstrates several desirable characteristics such as robustness in evidence retrieval performance, lower latency, etc. all across different document length ranges.

## 7 Limitations

- Although our approach demonstrated competitive performance in the information seeking setup, there is room for improvement, particularly when confronted with questions that require intricate multi-hop reasoning. Since we represent the document by summarizing each section, there is a potential loss of critical information that is essential for addressing complex queries necessitating multi-hop rea-

soning. Moving forward, we aim to explore methods that allow for accurate section selection while minimizing inference costs and mitigating information loss.

- Our experimental analyses primarily focus on enterprise-level language models, which require enterprise compute credits. In the future, we plan to explore the capabilities of more advanced open-source models as they become available, which may offer enhanced performance and accessibility.

- While our experiments have primarily centered around single-document use cases, we have yet to delve into the realm of retrieval involving multiple documents or collections. This area remains unexplored, and we anticipate investigating strategies and techniques to effectively handle such scenarios.

- Although our evaluations provided insight into how various summarizers impacted the final downstream performance, the current study did not inherently assess the quality of summarization. In future work, we aim to assess the summarization's faithfulness to the original content and its impact on end-to-end performance.

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
