# OpenReview forum: "Drilling Down into the Discourse Structure with LLMs for Long Document Question Answering"
_EMNLP/2023/Conference — EMNLP 2023 Findings_

### Official Review · Reviewer_nCeH · 2023-08-03

**Typos Grammar Style And Presentation Improvements:** n.a.
**Soundness:** 3

**Excitement:**

4: Strong: This paper deepens the understanding of some phenomenon or lowers the barriers to an existing research direction.

**Missing References:**

- https://arxiv.org/abs/2202.06991
- https://arxiv.org/pdf/2204.10628

**Paper Topic And Main Contributions:**

The study discusses the assessment of large language models (LLMs) for zero-shot long document evidence retrieval. LLMs' limited context length and high computational costs when processing large input sets are acknowledged as challenges. To overcome these issues, the authors propose techniques that exploit discourse structure in documents, leading to a condensed representation for a better understanding of relationships between different parts. Despite the reduction in token usage, the proposed approach retains almost all of the best zero-shot performance. Furthermore, the combination of their approach with a self-ask reasoning agent achieves excellent results in complex multi-hop question answering, coming close to the performance with gold evidence.

**Reasons To Accept:**

- The paper tackles an interesting problem, using LLM for fine-grained evidence retrieval. Compared to other baselines (exploiting LLM), the proposed method is effective in computational cost while maintaining the SOTA performance.

- The authors compare their method to many other baselines. The main experimental results seem significant (accuracy, computational cost (tokens processed, API call)). Furthermore, the ablation study conducted in 5.3.1 would be helpful to readers for engineering.

- The paper is well-organized, demonstrating figures and examples that help readers follow the paper.

**Reasons To Reject:**

- While the paper focuses on evidence paragraph selection, I think the scalability of the proposed method seems a bit limited. For example, how can the method be applied to an open-domain QA setup? It's not essential, but I would like to see a discussion regarding adapting the method to an open-domain setup (where the retrieval candidates are very large). Some reference papers regarding the issue are written in the "Missing References" section.

- minor) It would be nice if the authors can highlight the main difference of the proposed method compared to baselines in Section 4. (summary and filtering method)

**Reproducibility:**

4: Could mostly reproduce the results, but there may be some variation because of sample variance or minor variations in their interpretation of the protocol or method.

**Reviewer Confidence:**

4: Quite sure. I tried to check the important points carefully. It's unlikely, though conceivable, that I missed something that should affect my ratings.

---

> ### Author Rebuttal · Authors · 2023-08-29
>
> We thank the reviewer for their comments and positive feedback. In the following response, your comments are first stated and then followed by our point-by-point responses.
> > While the paper focuses on evidence paragraph selection, I think the scalability of the proposed method seems a bit limited. For example, how can the method be applied to an open-domain QA setup? It's not essential, but I would like to see a discussion regarding adapting the method to an open-domain setup (where the retrieval candidates are very large). Some reference papers regarding the issue are written in the "Missing References" section.
>
> >Missing References:
> https://arxiv.org/abs/2202.06991
> https://arxiv.org/pdf/2204.10628
>
> The strategy to represent a document section by a summary to determine its potential relevance to a question can also be leveraged for open domain question answering (ODQA). Typical approaches for ODQA use an efficient retriever to get a high recall set and then a more compute intensive re-ranker to get the relevant documents from the high recall set. We can use our method as a re-ranker in this pipeline where a summary would be computed for each document returned by the retriever. Then this summary can be used for identifying potentially relevant documents. Thereafter, akin to D3, we will use the entire document content of the filtered set to identify the relevant paragraphs. We can attempt a limited discussion of such an approach in this work in the camera ready version upon acceptance, but a full exploration of open domain QA is beyond the current scope and will be looked at in a future work (and we thank the reviewer for their suggestion!)
>
> We thank the reviewer for pointing out missing references, we will be adding them in the final version of the paper, if our paper gets accepted.
>
> > minor) It would be nice if the authors can highlight the main difference of the proposed method compared to baselines in Section 4. (summary and filtering method)
>
> The main difference of the proposed method (D3-base) with the baselines mentioned in Section 4 is the use of discourse structure. All of the baselines retrieve relevant paragraphs by only looking at ‘local information’ i.e only looking at the paragraphs/document chunks present in one inference call. The proposed method D3-base creates a condensed representation of the document where each section is represented by a section summary. This condensed representation allows us to pass in a global view of the document to the LLM, enabling the LLM to make accurate inferences about the sections that need to be selected. We then perform fine grained retrieval and choose the relevant paragraphs from selected sections for answering the question.
>
> Main differences:
> 1. Condensed representation of document presents global view of document to the LLM
> 2. Coarse retrieval (section selection) followed by fine-grained retrieval (paragraph selection) exploits discourse structure, reduces tokens processed and reduces latency.
>
> We will elaborate more on this comparison in the final version of the paper if our work gets accepted to the proceedings.

---

### Official Review · Reviewer_yp1i · 2023-08-03

**Soundness:** 3

**Excitement:**

3: Ambivalent: It has merits (e.g., it reports state-of-the-art results, the idea is nice), but there are key weaknesses (e.g., it describes incremental work), and it can significantly benefit from another round of revision. However, I won't object to accepting it if my co-reviewers champion it.

**Paper Topic And Main Contributions:**

The paper introduces a new method for the task of evidence retrieval in long document question answering. The main challenge in dealing with long documents is that their size usually surpasses the context size of Transformer-based language models. To tackle this challenge, the proposed method relies on the document structure by breaking down long documents into their constituent sections. The hierarchical structure of sections would be flattened by concatenating the parent section titles into their respective subsections. The method undertakes the task in three main steps: summarizing sections, finding relevant sections to an input question, and finally finding relevant paragraphs from relevant sections given a question. A key contribution of the proposed method is that it works in a zero-shot fashion with no labelled data requirements.

Specifically, each section is first summarized using BART$_\textrm{large}$, fine-tuned on the CNN/DailyNews corpus. In the next step, given a question and a list of summarized sections, relevant sections are selected by eliciting a prompt from an LLM. Subsequently, using a similar method to the previous step, relevant paragraphs are determined by instructing an LLM to find relevance based on an input question and relevant sections that are split into paragraphs or contiguous chunks of text.

The method is tested on two benchmarks: QA over research papers (QASPER) and multi-hop QA (HotpotQA). The method is compared against its different variants as well as baselines from previous work. The paper also considers the number of processed tokens and the number of API calls to measure the cost of using LLMs. The results show that the proposed method achieves competitive performance on both tasks.

**Questions For The Authors:**

One of the baselines, map-reduce that directly finds relevant paragraphs given a question (akin to the proposed method but excluding the first two steps), is implemented using the LangChain library, but it is specified (L443-L454) that the implementation uses a “suboptimal few-shot prompting”. Why is this implementation not consistent with other baselines? The idea seems straightforward for implementing without LangChain.

**Reasons To Accept:**

1. QA over long documents poses challenges that Transformer models tend to struggle with, and the techniques introduced in this paper can shed light on how to leverage document structure to address such challenges.
2. The proposed method is simple and effective, making it easy to adopt in the community.

**Reasons To Reject:**

1. The summaries generated in the first step were not assessed to measure their impact on the overall performance. We can see small improvements when BART$_\text{large}$ is replaced with GPT-3 or Vicuna. Questions including are the summaries mostly extractive or what happens if the summary is hallucinated or how the quality of summaries varies across different sections are worth a discussion in the paper.

2. It is hard to figure out where the proposed method makes mistakes which can be essential for future work. An error analysis would strengthen the paper to highlight its limitations.

3. One assumption of this work is the presence of sections in the documents. Many long documents such as news articles are not usually organized into sections. However, leveraging sections seems to play an important role in the proposed method. In Table 1, for instance, the difference between the evidence F$_1$ of “chunk” and “base” is around 14%. On the other hand, the “chunk” baseline is more general in that it can be used without sections as well. It actually works well in multi-hop QA (Table 2), which raises the question whether the use of discourse as claimed in the paper is really impactful in the task.

4. The proposed method does not work well in multi-hop QA. On HotpotQA (Table 2), using a document ranking model (MonoT5) works better than the proposed method and its variants (20-30% lead). Even after applying self-ask, the gap is 4-6%, thus telling us simple methods are still useful.

5. Details about how documents are split when the length of a prompt exceeds the LLM maximum context size is missing. In the paper, it is stated that paragraphs would be contiguously fragmented to fit into the context size (L305-L308). Which paragraph would be fragmented? One option is to evenly fragment all paragraphs and another option is to fragment the last paragraph. In any way, the prompt would be expanded to multiple prompts. Now, the newly created prompts may include chopped paragraphs. Would this affect the performance? Also, would other paragraphs that were not fragmented be repeated in the subsequent prompts?

6. In Table 1, evidence retrieval of the “paragraph” and “chunk” baselines are low (32.02 and 35.59), but their answer effectiveness (it is not explained what “answering performance” actually means) 47.92 and 49.00 becomes close to the best model (50.13). This observation needs an explanation.

**Reproducibility:**

4: Could mostly reproduce the results, but there may be some variation because of sample variance or minor variations in their interpretation of the protocol or method.

**Reviewer Confidence:**

4: Quite sure. I tried to check the important points carefully. It's unlikely, though conceivable, that I missed something that should affect my ratings.

**Typos Grammar Style And Presentation Improvements:**

- Tables 1 and 2 provide “answer performance”, but how final answers are obtained after retrieving evidence from the document is only visualized in the appendix. It’d be better to clarify this in the text as well.
- L163, allows realizing $\rightarrow$ allows for…
- L382, ito $\rightarrow$ to
- When making references to tables and figures, it is recommended to use capitalization, e.g. L484, L567.
- Sections 5.3.2 and 5.3.4 do not contain any references to Figure 2 and Table 4, respectively.

---

> ### Author Rebuttal · Authors · 2023-08-29
>
> Thank you for the detailed review!  In the following response, the reviewer questions/comments are first stated, followed by our point-by-point responses.
>
> > One of the baselines, map-reduce that directly finds relevant paragraphs given a question (akin to the proposed method but excluding the first two steps), is implemented using the LangChain library, but it is specified (L443-L454) that the implementation uses a “suboptimal few-shot prompting”. Why is this implementation not consistent with other baselines? The idea seems straightforward for implementing without LangChain.
>
> * We used the LangChain library because it  has a standard, well-tested and widely used implementation of map-reduce. We noticed that the standard prompt used in Langchain for the reduce stage did not perform well. This is because of the incompatibility of the prompt with the target domain as it had 2 in context examples one from the contract domain and one from a transcript of a speech. We replace the combine_prompt with a zero-shot prompt for the map-reduce optimized baseline.
> * We have further optimized the map-reduce implementation by processing the input in chunks rather than paragraphs as mentioned in L383-L392. We have reported the scores for this as map-reduce-optimized in Table 1.
>
> > The summaries generated in the first step were not assessed to measure their impact on the overall performance. We can see small improvements when BART is replaced with GPT-3 or Vicuna. Questions including are the summaries mostly extractive or what happens if the summary is hallucinated or how the quality of summaries varies across different sections are worth a discussion in the paper.
>
>  * We agree with the reviewer that the quality of summarization plays an important role in the downstream performance. While different summarizers were extrinsically evaluated in terms of final evidence retrieval performance in table 3, we would include an additional sub-section devoted towards more rigorous evaluation of the summarization quality. Particularly, we would be using reference-free evaluation metrics (such as Semantic Distribution Correlation which generate a document both with and without a summary as a prompt, and then computes the difference in information content between two generated documents for metric computation [Reference-free Summarization Evaluation via Semantic Correlation and Compression Ratio by Liu et. al. 2022]) and measure its correlation with the end-to-end retrieval performance. Similarly, we intend to evaluate the summarization faithfulness with respect to the original document content and assess how it affects the retrieval performance. We would be including these analyses in our final-version, which won’t require much edits and can be conveniently presented in a sub-section, if our paper gets accepted to the proceedings.
>
> > It is hard to figure out where the proposed method makes mistakes which can be essential for future work. An error analysis would strengthen the paper to highlight its limitations.
>
> * In the table below, we compare the evidence retrieval performance with map-reduce optimized over different question categories in QASPER to identify differences in model performance. While our approach is precise in identifying the evidence paragraphs, it occasionally falls short in identifying all pertinent evidence (i.e lower recall). This indicates that representing a section with a summary leads to loss of information and the LLM may miscategorize its relevancy to a question due to this loss of information. Designing an effective strategy to prevent loss of vital information for a particular question may be a future research direction. We will include this discussion in our camera-ready version and include qualitative examples if our paper gets accepted and provide researchers with a clearer perspective on potential extensions of our work.  Also the categorization of the results below based on question type will help researchers in effectively understanding the question categories where our performance falls short. We will also add some qualitative error analysis in our final version.
>
>
> Table: Evidence Retrieval Performance over different question types in the QASPER dataset
>
> | Type of question | Evidence Precision [D3/map-reduce-optimized] | Evidence Recall  [D3/map-reduce-optimized]| Evidence F1  [D3/map-reduce-optimized] |
> | ---------------- | ------------------------------------------ | --------------------------------------- | ----------------------------------- |
> | Extractive       | 52.63 / 52.19                              | 71.9 / 81.49.                                             | 56.39 / 57.46                       |
> | Abstractive      | 41.25 / 45.1 | 52.51 / 71.57  | 41.78 / 50.12 |
> | Yes/No           | 43.6 / 38.79 | 59.6 / 55.31  | 45.53 / 40.18 |
> | Unanswerable |  42.94/ 25.74 | 44.4/25.46 | 43.43/25.10 |
>
>
>
> > Details about how documents are split when the length of a prompt exceeds the LLM maximum context size is missing. In the paper, it is stated that paragraphs would be contiguously fragmented to fit into the context size (L305-L308). Which paragraph would be fragmented? One option is to evenly fragment all paragraphs and another option is to fragment the last paragraph. In any way, the prompt would be expanded to multiple prompts. Now, the newly created prompts may include chopped paragraphs. Would this affect the performance? Also, would other paragraphs that were not fragmented be repeated in the subsequent prompts?
>
> * If the number of paragraphs in Pq exceeds the maximum context length of LLM, we make multiple LLM calls. In each call, we fit the maximum number of paragraphs that can fit into the context length, ensuring that paragraphs are not ‘chopped’. We will make this more clear in the camera-ready version if our paper gets accepted.
>
> > The proposed method does not work well in multi-hop QA. On HotpotQA (Table 2), using a document ranking model (MonoT5) works better than the proposed method and its variants (20-30% lead). Even after applying self-ask, the gap is 4-6%, thus telling us simple methods are still useful.
>
> * An explanation of the performance of the baselines and the proposed method is mentioned in L485-489.
> * To summarize, due to the method by which the HotpotQADoc dataset was constructed, only the paragraphs utilized for generating answers were preserved, which does not imply the irrelevance of other paragraphs. This is reflected by the fact that the Evidence F1 score does not correlate with the Answer F1 score in Table 2 even if the same question answering model is used. In the zero-shot set of experiments, MonoT5 achieves the highest evidence performance (62.33 evidence F1) but performs the same as Map reduce Optimized (39.52 Evidence F1) in Answer F1.
>
>
>
>
> > One assumption of this work is the presence of sections in the documents. Many long documents such as news articles are not usually organized into sections. However, leveraging sections seems to play an important role in the proposed method. In Table 1, for instance, the difference between the evidence F of “chunk” and “base” is around 14%. On the other hand, the “chunk” baseline is more general in that it can be used without sections as well. It actually works well in multi-hop QA (Table 2), which raises the question whether the use of discourse as claimed in the paper is really impactful in the task.
>
> > In Table 1, evidence retrieval of the “paragraph” and “chunk” baselines are low (32.02 and 35.59), but their answer effectiveness (it is not explained what “answering performance” actually means) 47.92 and 49.00 becomes close to the best model (50.13). This observation needs an explanation.
>
> 1. Performance of Chunk baseline and the competitive performance of Chunk / Paragraph baseline:
>     * We agree with the reviewer that the chunk baseline used in the paper is applicable in cases where there is no discourse structure. However, the chunk baseline is akin to a high-recall brute-force solution for retrieval with LLMs. It can perform reasonably well however at a high cost and latency.
>
> 2.  Performance of Chunk on HotpotQA dataset:
>         Please refer to the previous point in this response
>
> 3.  Fine grained analysis of Chunk / Paragraph vs D3-base:
>        * In the below table we have listed the Evidence Precision, Evidence Recall, Evidence F1, Average Number of paragraphs retrieved (per question), Tokens Processed and Number of API calls of the Chunk / Paragraph baseline and our method D3-base. (on QASPER Dataset) When processing paragraphs using the chunk method, the LLM does not have a global view of the document. It has access only to the paragraphs present in the context. This suboptimal processing devoid of global context results in the observation that the chunk / paragraph method tends to categorize a substantial number of paragraphs as relevant to the question, as shown in the table below. This is the reason why the chunk baseline has high recall but low precision. As the recall is higher, the chunk based methods have access to more paragraphs that contain the ground truth answer and hence we can observe competitive answering performance.
>      * When employing a retrieval mechanism for LDQA, we would want the retriever to have both high precision and high recall, while minimising cost. Compared to the chunk baseline, our method D3-base has more balanced precision recall values while processing only 36% of the tokens.
>
> Table: Fine grained analysis of evidence performance using precision, recall, F1 and average number of paragraphs retrieved (per question).
> (QASPER dataset)
> | Method    | Evidence Precision | Evidence Recall | Evidence F1 | Average Number of paragraphs retrieved (per question) | Tokens Processed | Number of API calls |
> | --------- | ------------------ | --------------- | ----------- | ------------------------------------------------------- | ---------------- | ------------------- |
> | Gold      | \-                 | \-              | \-          | 1.48                                                   | \-               | \-                  |
> | Chunk     | 27.82              | 71.96           | 35.59       | 6.72                                                  | 5411.44          | 2.44                |
> | Paragraph | 24.53              | 74.96           | 32.02        | 8.47                                                   | 8519.37          | 47.24               |
> | D3-base   | 47.53              | 62.43           | 49.92      | 2.92   |  1980.94 | 1.99
>
> > We also thank the reviewer for highlighting Presentation Improvements, we will make the corrections in the final version

---

### Official Review · Reviewer_SsPB · 2023-08-08

**Soundness:** 3

**Excitement:**

3: Ambivalent: It has merits (e.g., it reports state-of-the-art results, the idea is nice), but there are key weaknesses (e.g., it describes incremental work), and it can significantly benefit from another round of revision. However, I won't object to accepting it if my co-reviewers champion it.

**Paper Topic And Main Contributions:**

This work proposes a suite of techniques to the summarize document paragraphs to create a condensed representation of the document for a particular question using local (and less expensive) LLMs. The condensed paragraphs are then fed to OpenAI based LLMs to filter out irrelevant paragraphs from which the OpenAI based LLM can generate the final answer to the question. The paper presents evaluation showing that the proposed technique retain 99.6% of the best zero-shot approach’s performance, while requiring to process only 26% of the total documents tokens via the expensive OpenAI LLMs.

**Reasons To Accept:**

This work proposes a suite of techniques to the summarize document paragraphs to create a condensed representation of the document for a particular question using local (and less expensive) LLMs. The condensed paragraphs are then fed to OpenAI based LLMs to filter out irrelevant paragraphs from which the OpenAI based LLM can generate the final answer to the question. The paper presents evaluation showing that the proposed technique retain 99.6% of the best zero-shot approach’s performance, while requiring to process only 26% of the total documents tokens via the expensive OpenAI LLMs.

**Reasons To Reject:**

Limited Impact:
1.  The proposed technique still needs to process all the tokens in the document via the local LLMs to condense the document content before making calls to the OpenAI LLM for additional processing. The impact of the proposed technique might be better understood if the authors presented comparisons based on end-to-end processing times of the various systems.
2. In the current environment, where new LLMs are constantly released (whether open sourced or via proprietary APIs) with larger contexts and response time SLAs, it is hard to evaluate relevance of this work in the near future.

Limited Novelty: The authors propose a technique that summarizes a document using local LLMs to reduce the document that needs to be processed via the expensive OpenAI LLMs. In my opinion, this will have limited interest to the main conference audience and is more appropriate for workshop.

**Reproducibility:**

3: Could reproduce the results with some difficulty. The settings of parameters are underspecified or subjectively determined; the training/evaluation data are not widely available.

**Reviewer Confidence:**

4: Quite sure. I tried to check the important points carefully. It's unlikely, though conceivable, that I missed something that should affect my ratings.

---

> ### Author Rebuttal · Authors · 2023-08-29
>
> We thank the reviewer for their comments and their feedback. In the following response, the comments are first stated and then followed by our point-by-point response
>
> > The proposed technique still needs to process all the tokens in the document via the local LLMs to condense the document content before making calls to the OpenAI LLM for additional processing. The impact of the proposed technique might be better understood if the authors presented comparisons based on end-to-end processing times of the various systems.
>
> We would like to point out that the processing of the document to derive the condensed representation is done using a lightweight fine-tuned transformer model and thus this computation can be done in a fraction of a second. Moreover, once we get the condensed representation, we can answer all the questions pertaining to that document at once, thereby making the effective processing time per document even smaller. Since our observation revealed that the primary time consumption is attributed to LLM usage, we've reported the number of sequential API calls, reflecting latency. This metric has two benefits: firstly, it's deterministic, and secondly, for different LLMs, multiplying the API call count by their average turnaround time provides a quick approximation of the evidence retrieval time.
> That being said, we agree with your point that the computation of the end-to-end processing time can better communicate the results. We computed the end-to-end processing time of the LLM based systems (which used gpt-3.5-turbo) in Table 1 (The langchain implementation of Map-reduce takes very long as the API calls ask LLMs to generate a longer sequence resulting in more latency). As we see in the following table, **our approach is significantly more time and token-efficient**. We will incorporate this change in the camera-ready version if our paper gets accepted to the proceedings.
>
> Table:  End-to-End processing time (per question) of LLM based systems on the QASPER dataset
>
> |Method              |Overall Answer F1|Evidence F1|Number of Tokens processed|Number of API Calls|Average Time (per question in seconds)|
> |--------------------|-----------------|-----------|--------------------------|-------------------|----------------------------------------|
> |Paragraph           |47.92            |32.03      |8519.37                   |47.24              |3.48                                    |
> |Chunk               |49.00            |35.59      |5411.44                   |2.44               |2.49                                    |
> |Map-reduce          |39.26            |12.84      |12730.13                  |48.24              |33.82                                   |
> |Map-reduce-optimized|50.13            |50.11      |7491.97                   |3.69               |3.59                                    |
> |D3 -BASE            |47.45            |49.92      |1980.94                   |1.99               |1.44                                    |
>
>
>
> > In the current environment, where new LLMs are constantly released (whether open sourced or via proprietary APIs) with larger contexts and response time SLAs, it is hard to evaluate relevance of this work in the near future.
>
> LLMs are increasingly being adopted for document based question answering. Many prior approaches for LDQA using LLMs do not consider crucial aspects such as latency, and compute cost and have primarily focused on achieving high performance in terms of accuracy and effectiveness. Our work makes a novel contribution in addressing the real-world practicalities involved in using LLMs for LDQA by finding efficient ways to save computational / monetary costs. If our paper gets accepted to the conference proceedings, the significance of cost-reduction techniques can reach a broader audience. This would subsequently foster increased attention of the researchers towards this problem, thereby making LLMs more applicable for document based question answering. Adapting our proposed algorithm will also give engineers   cushioning to apply LLMs without worrying much on cost and infrastructures.

---

### Official Review · Reviewer_4fXv · 2023-08-12

**Soundness:** 3

**Excitement:**

4: Strong: This paper deepens the understanding of some phenomenon or lowers the barriers to an existing research direction.

**Missing References:**

As per my knowledge, the current SOTA for the QASPER dataset is COLT5-XL with an F-1 score of 53.9.
COLT5: Faster Long-Range Transformers with Conditional Computation
https://arxiv.org/pdf/2303.09752v2.pdf

**Paper Topic And Main Contributions:**

This paper introduces a method called D3 (Drilling Down into the Discourse) for long document question answering (LDQA) which uses the inherent discourse structure in long documents to create a condensed representation that enables more efficient processing. In D3 the document is divided into sections and each section is summarized. The section names and summaries are provided as context to a large language model (LLM), which identifies relevant sections. Then the paragraphs in the relevant sections are processed to retrieve fine-grained evidence. This allows focusing on pertinent portions of the document while capturing global context. Several zero-shot strategies are explored for evidence retrieval, including using the LLM directly or a re-ranker like MonoT5. The approach is evaluated on QASPER and HotpotQA and shows competitive performance while being computationally cheaper.


**Questions For The Authors:**

Question A) Do you have results on using simpler baselines such as DPR to retrieve relevant chunks and pass into the LM ?

**Reasons To Accept:**

The paper proposes a systematic way to leverage discourse structure for more efficient evidence retrieval. It shows Competitive zero-shot performance while using significantly fewer tokens. It also seems to be easily generalizable by replacing QA LLM, summarizer etc.

**Reasons To Reject:**

I believe the paper is missing some even primitive baselines such as using DPR to find top-k relevant chunks and feeding them into the Language Model rather than prompting the LM for what passages are important.

**Reproducibility:**

4: Could mostly reproduce the results, but there may be some variation because of sample variance or minor variations in their interpretation of the protocol or method.

**Reviewer Confidence:**

4: Quite sure. I tried to check the important points carefully. It's unlikely, though conceivable, that I missed something that should affect my ratings.

---

> ### Author Rebuttal · Authors · 2023-08-29
>
> We thank the reviewer for their comments and their positive feedback. In the following response, the comments are first stated and then followed by our point-by-point response
>
> > Question A) Do you have results on using simpler baselines such as DPR to retrieve relevant chunks and pass into the LM ?
>
> > I believe the paper is missing some even primitive baselines such as using DPR to find top-k relevant chunks and feeding them into the Language Model rather than prompting the LM for what passages are important.
>
> For open domain question answering (ODQA), a standard pipeline uses a retriever (such as BM25 or DPR) to retrieve top-100 relevant passages from millions of candidates. These top-100 passages are then re-ranked again using methods like monoT5 to send the most relevant passages to the reader. In the long document question answering (LDQA) scenario, the retrieval happens over a much limited set of candidates (100 passages in total in documents) similar to the reranking step. Following this intuition we have compared our method with MonoT5, a reranking algorithm in the paper.
>
> We had initially run experiments with simpler models (such as DPR) trained for the objective of semantic search. We first ran some experiments to determine the value of top_k that gives best performance for these methods (DPR, all-MiniLM-L6-v2) and found the ideal top_k=3. We then applied them for retrieval on QASPER dataset. The results of these experiments are tabulated below. As we can see, monoT5 achieved the best Evidence F1 performance, far higher than simpler baselines such as DPR. Hence we compared mainly with monoT5 in our paper (when comparing our method to embedding-based retrieval).
>
> For comprehensive analysis, we are including the results of these experiments below:
>
> Table: Additional experiments on QASPER dataset using semantic search methods, monoT5, DPR
>
> |Method| Extractive F1 (Answering performance) | Abstractive F1 (Answering performance) | Yes/No F1 (Answering performance)| Unanswerable F1 (Answering performance) |Overall F1 (Answering performance) | Evidence F1
> |-------------------------------------------------|-----------|-----------|-----------|-----------|-----|-----------|
> |monoT5                                           |42.84      |25.84      |82.23      |69.09      |47.21|34.23      |
> |DPR (model trained on combination of multiple datasets)         |31.58      |18.57      |78.46      |84.33      |42.11|19.32      |
> |all-MiniLM-L6-v2   |34.28      |22.94      |69.67      |75.79      |41.03|24.89      |
> |msmarco-bert-base-dot-v5| 37.48      |26.47      |80.69      |73.06      |44.84|25.85      |
> |multi-qa-mpnet-base-cos-v1 |34.94| 23.72| 74.26 | 69.08 | 41.70 |26.38 |
> |cross-encoder-ms-marco-MiniLM-L-12-v2  |38.69      |23.25      |78.04      |71.42      |43.48|30.76      |
>
> > As per my knowledge, the current SOTA for the QASPER dataset is COLT5-XL with an F-1 score of 53.9. COLT5: Faster Long-Range Transformers with Conditional Computation https://arxiv.org/pdf/2303.09752v2.pdf
>
> We thank the reviewer for pointing out missing references, we will include them along with the above table in the camera-ready version of the paper if our paper gets accepted.

---

### Meta-Review · Area_Chair_LLVs · 2023-09-17

**Recommendation:** 4

**Metareview:**

The reviewers agree this work proposes an interesting and useful new method for retrieval/QA from long documents, promising significant token efficiency, without sacrificing overall accuracy. The authors’ commitments to updating the paper with new results and analysis, requested by reviewers, will improve the applicability and clarity of this method to practitioners.

---

### Decision · Program_Chairs · 2023-10-07

**Decision:**

Accept-Findings

**Comment:**

The reviewers agree this work proposes an interesting and useful new method for retrieval/QA from long documents, promising significant token efficiency, without sacrificing overall accuracy. The authors’ commitments to updating the paper with new results and analysis, requested by reviewers, will improve the applicability and clarity of this method to practitioners.